# Effect of the Combination of Creatine Monohydrate Plus HMB Supplementation on Sports Performance, Body Composition, Markers of Muscle Damage and Hormone Status: A Systematic Review

**DOI:** 10.3390/nu11102528

**Published:** 2019-10-20

**Authors:** Julen Fernández-Landa, Julio Calleja-González, Patxi León-Guereño, Alberto Caballero-García, Alfredo Córdova, Juan Mielgo-Ayuso

**Affiliations:** 1Laboratory of Human Performance, Department of Physical Education and Sport, Faculty of Education, Sport Section, University of the Basque Country, 01007 Vitoria, Spain; julenfdl@hotmail.com (J.F.-L.); julio.calleja.gonzalez@gmail.com (J.C.-G.); 2Faculty of Psychology and Education, University of Deusto, Campus of Donostia-San Sebastián, 20012 San Sebastián, Spain; patxi.leon@deusto.es; 3Department of Anatomy and Radiology, Faculty of Health Sciences, University of Valladolid. Campus de Soria, 42003 Soria, Spain; albcab@ah.uva.es; 4Department of Biochemistry, Molecular Biology and Physiology, Faculty of Health Sciences, University of Valladolid, 42003 Soria, Spain; a.cordova@bio.uva.es

**Keywords:** sport nutrition, anaerobic, aerobic, body composition, muscle recovery

## Abstract

Although there are many studies showing the isolated effect of creatine monohydrate (CrM) and β-hydroxy β-methylbutyrate (HMB), it is not clear what effect they have when they are combined. The main purpose of this systematic review was to determine the efficacy of mixing CrM plus HMB in comparison with their isolated effects on sports performance, body composition, exercise induced markers of muscle damage, and anabolic-catabolic hormones. This systematic review was carried out in accordance with PRISMA (Preferred Reporting Items for Systematic Reviews and Meta-Analyses) statement guidelines and the PICOS model, for the definition of the inclusion criteria. Studies were found by searching PubMed/MEDLINE, Web of Science (WOS), and Scopus electronic databases from inception to July 3rd 2019. Methodological quality and risk of bias were assessed by two authors independently, and disagreements were resolved by third-party evaluation, in accordance with the Cochrane Collaboration Guidelines samples. The literature was examined regarding the effects of the combination of CrM plus HMB on sport performance using several outcome variables (athletic performance, body composition, markers of muscle damage, and hormone status). This systematic review included six articles that investigated the effects of CrM plus HMB on sport performance (two on strength performance, showing improvements in one of them; three on anaerobic performance, presenting enhancements in two of them; and one on aerobic performance, not presenting improvements), body composition (three on body mass, showing improvements in one of them; two on fat free mass, presenting increases in one of them; and two on fat mass, showing decreases in one of them) and markers of muscle damage and hormone status (four on markers of muscle damage and one on anabolic-catabolic hormones, not showing benefits in any of them). In summary, the combination of 3–10 g/day of CrM plus 3 g/day of HMB for 1–6 weeks could produce potential positive effects on sport performance (strength and anaerobic performance) and for 4 weeks on body composition (increasing fat free mass and decreasing fat mass). However, this combination seems to not show positive effects relating to markers of exercise-induced muscle damage and anabolic-catabolic hormones.

## 1. Introduction

Supplements and sport foods may help to prevent or treat nutrition deficiencies, and occasionally have a direct ergogenic effect [1]. However, there are few supplements supported by strong evidence that produce a significant effect on sports performance [1]. In this sense, while there are several supplements that have strong scientific evidence for use in sports-specific situations using evidence-based protocols, such as creatine monohydrate (CrM), others—although deserving of further research—require more scientific support; for example, β-hydroxy β-methylbutyrate (HMB) [2].

CrM is one of the most popular performance supplements used by athletes that may promote aerobic [3,4] and anaerobic performance [5], strength [6,7,8,9], body composition [6,8,10], reduced markers of exercise induced muscle damage, and anabolic-catabolic hormones [11,12]. The main action of CrM is to increase the muscle creatine (CR) stores to replace adenosine triphosphate (ATP) degradation during exercise [13]. Moreover, it increases the muscle glycogen pool by stimulating muscle glycogen synthesis based on the augmentation of muscle cells and muscle CR content [14,15]. Additionally, this effect could increase creatine-phosphocreatine (Cr-PCr) shuttling, improving aerobic capacity [16]. Furthermore, CR supplementation increases lean tissue mass and upper and lower body muscular strength [17].

HMB could also enhance sports performance in terms of aerobic power and capacity [18,19,20], anaerobic capacity [18,20], strength [21,22,23,24], body composition [18,25], markers of muscle damage [26], and hormone status [25]. The main role of HMB is to stimulate muscle protein synthesis by an up-regulation of Mammalian Target of Rapamycin kinase (mTOR) [27]. Additionally, HMB could augment muscle glycogen storage [28,29] and can increase gene expression of peroxisome proliferator-activated receptor gamma co-activator 1-alpha α (PGC-1α), enhancing mitochondrial biogenesis, and hence oxidative function, to enhance aerobic capacity [30]. Moreover, this benefit in sports performance could be motivated by an augmented lean body mass (LBM) and/or fat free mass (FFM) [20,22,31], and reduced fat mass (FM) [18,20,31].

Given that these two supplements have different physiological pathways to improve performance [13,14,15,28,29,32,33,34], it could be assumed that the combination of both complements would improve sports performance compared to taking them alone. Therefore, some authors have considered the utilization of both supplements together (CrM plus HMB) with the aim of producing an additive or synergistic effect. To the best of the authors’ knowledge, the results of the studies investigating combined supplementation are not clear. Some studies show possible improvements in performance [35,36,37], body composition (increases FFM and decreases FM) [36,37], markers of muscle damage [36], and hormones status [38], but others found no changes in these outcomes [38,39,40]. Therefore, the main purpose of this systematic review was to determine the efficacy of mixing CrM plus HMB in improving sports performance, body composition (increases FFM, LBM, and decreases FM), markers of exercise induced muscle damage, and anabolic-catabolic hormones in comparison with their isolated effects.

## 2. Methods

### 2.1. Literature Search Strategies

This systematic review was carried out in accordance with PRISMA^®^ (Preferred Reporting Items for Systematic Reviews and Meta-Analyses) statement guidelines and the PICOS model for the definition of the inclusion criteria: P (Population): “athletes”, I (Intervention): “impact of the combination of HMB and CrM in sport”, C (Comparators): “same conditions with control, placebo, only HMB or only CrM”, O (Outcome): “sport performance, body composition, markers of muscle damage and hormone status”, and S (study design): “clinical trial” [41]. A systematic search of the current scientific literature was undertaken for studies that investigated the mixed supplementation of CrM plus HMB in sports performance and recovery. Studies were found by searching PubMed/MEDLINE, Web of Science (WOS), and Scopus from inception to July 3rd 2019, using the following Boolean search equation: (“creatine monohydrate supplementation”[All Fields] OR “creatine supplementation”[All Fields]) AND (”HMB supplementation”[All Fields] OR ”beta hydroxy beta methylbutyrate supplementation”[All Fields] OR (beta-Hydroxy[All Fields] AND methylbutyrate[All Fields] AND supplementation[All Fields])) AND (”muscle damage” [All Fields] OR “hormone status”[All Fields] OR (”athletes”[MeSH Terms] OR ”athletes”[All Fields]) OR (”exercise”[MeSH Terms] OR ”exercise”[All Fields]) OR ”sport performance”[All Fields] OR ”body composition”[All Fields]). Through this equation, relevant articles in this field were obtained by applying the snowball strategy. All titles and abstracts from the search were cross-referenced to identify duplicates and any potential missing studies. The titles and abstracts were screened for a subsequent full-text review. The search for published studies was independently performed by two authors (J.F.L. and J.M.A.), and disagreements about all outcomes were resolved through discussion.

### 2.2. Inclusion and Exclusion Criteria

For the articles obtained in this search, the following inclusion criteria were applied to select studies: (1) a well-designed experiment that included ingestion of the combination of CrM plus HMB; (2) with an identical experimental situation related to the ingestion of a placebo, CrM only, and/or HMB only; (3) testing the effects of mixed supplementation on sports performance, body composition, markers of muscle damage, and/or hormone status; (4) clinical trial; (5) with clear information regarding the administration of ergogenic aids (dosage and timing); and (6) published in any language. The following exclusion criteria were applied to the experimental protocols of the investigation: (1) supplementation was mixed with other supplements or was a multi-ingredient compound; (2) carried out in participants with a previous condition, injury, or health problems. There were no filters applied to the athletes’ level, gender, ethnicity, or age to increase the analytic power of the analysis.

Once the inclusion/exclusion criteria were applied to each study, data on study source (including authors and year of publication), study design, supplement administration (dose and timing), sample size, characteristics of the participants (level, race and gender), and final outcomes of the interventions were extracted independently by two authors (J.F.L. and J.M.A.) using a spreadsheet (Microsoft Inc, Seattle, WA, USA). Subsequently, disagreements were resolved through discussion until a consensus was reached, or by third-party adjudication (J.C.G.).

### 2.3. Study Selection

One reviewer (J.F.L.) searched the databases and selected the studies. A second reviewer (J.M.A.) was available to help with study eligibility. No disagreements about the appropriateness of an article were encountered.

### 2.4. Outcome Measures

The literature was examined regarding the effects of the combination of CrM plus HMB in sports performance using several outcome variables, such as athletic performance [35,36,37,39,40], body composition [36,37,40], markers of muscle damage [35,36,37,38,39], and hormone status [38].

### 2.5. Quality Assessment of the Experiments

Methodological quality and risk of bias were assessed by two authors independently (J.F.L. and J.M.A.), and disagreements were resolved by third-party evaluation (J.C.G.), in accordance with the Cochrane Collaboration Guidelines [42]. The items on the list were divided into six domains: selection bias (random sequence generation, allocation concealment); performance bias (blinding of participants and researchers); detection bias (blinding of outcome assessment); attrition bias (incomplete outcome data); reporting bias (selective reporting); and other types of bias. For each research paper, domains were judged by consensus (J.F.L., J.M.A.), or third-party adjudication (J.C.G.). They were characterized as ‘low’ if criteria for a low risk of bias were met (plausible bias unlikely to seriously alter the results) or ‘high’ if criteria for a high risk of bias were met (plausible bias that seriously weakens confidence in the results), or it was considered ‘unclear’ (plausible bias that raises some doubt about the results), if the risk of bias was unknown. Full details are given in Figure 1 and Figure 2.

## 3. Results

### 3.1. Main Search

A search of electronic databases revealed seventeen relevant studies, with an additional three studies found by searches in reference lists (Figure 3). After removing duplicate studies (*n* = 14) and screening titles and abstracts (*n* = 6), eight studies were retained [35,36,37,38,39,40,43,44]. Following full-text screening, only two studies were excluded [43,44] (one was combined with a non-usual training [43] and the other one was performed with only one of the supplements [44]). Thereby, six studies were included for this systematic review [35,36,37,38,39,40].

The design of the six studies included one randomized, double-blind, placebo-controlled study [36], two randomized, placebo-controlled studies [35,37], and three controlled studies [38,39,40]. Five out of six studies were conducted in intermittent team sports: three in rugby [38,39,40], one in basketball [37], and one in soccer [35]. The last study was carried out on healthy males [36]. The sum of all study participants included in this review were 201 males, with 161 being participants in high-level sports leagues (five studies) [35,37,38,39,40] and the remaining 40 being moderately trained participants [36].

Some studies divided the participants into four different supplementation groups: placebo group (PLAG) or control group (CON), creatine group (CrMG), HMB group (HMBG), and CrM plus HMB group (CrM/HMBG) [36,37]. Other studies assigned participants to three groups: PLAG or CON, HMBG, and CrM/HMBG [35,38,39,40].

### 3.2. CrM and HMB Supplementation

The duration of the interventions in the studies was between six days and six weeks (42 days). In four of the studies included in this systematic review the supplementation protocol consisted of 3 g/day of CrM and 3 g/day of HMB [35,38,39,40]. The exceptions were the studies by Jowko et al. [36] and Zajac et al. [37], in which the CrM supplementation consisted of 20 g/day for the first 7 days followed by 10 g/day the next 14 days, and 15 g/day the first 5 days followed by 5 g/day the next 25 days, respectively. Moreover, the training program frequency performed by the athletes was 3 or 4 days per week [36,37,38,39,40] or 6 days per week [35] in a resistance training program.

### 3.3. Sports Performance Outcomes

Table 1 presents the different tests carried out to determine the performance measured outcomes by running an anaerobic speed test (RAST) [35], one repetition maximum (1-RM) test of different strength exercises [36], multistage aerobic capacity test, cycling maximally for 60 seconds [39], muscular strength, muscular endurance, leg power tests [40], and a triple Wingate test [37].

The combination of CrM plus HMB showed improvements in strength performance in one study [36], but the other study that measured this parameter did not find performance changes among groups [40]. On the other hand, anaerobic capacity was enhanced in two investigations [36,37] when the athletes ingested CrM plus HMB, although the study by O’Connor et al. [39] did not find any improvement. The last performance variable was aerobic capacity, measured by a multistage aerobic capacity test, and it was unchanged in all groups [39].

### 3.4. Body Composition Outcomes

Table 2 shows the body composition measures. Zajac et al. [37] found reductions in FM (regarding CON/PLG and CrMG) in basketball players when they were supplemented with CrM plus HMB, although Jowko et al. [36] did not. On the other hand, Zajac et al. found an increase in BM and FFM (regarding CON/PLG and HMBG), contrasting to studies by Jowko et al. [36] and O’Connor et al. [40], who did not find changes in this parameter among groups when CrM plus HMB was ingested.

### 3.5. Markers of Muscle Damage and Hormone Status Outcomes

Table 3 displays muscular blood isoenzymes, such as creatine kinase (CK) [35,36,38], lactate dehydrogenase (LDH) [35], and blood lactate concentration (LA) [37,39], which were unchanged after supplementation with CrM plus HMB. Moreover, anabolic/catabolic blood hormones (testosterone and cortisol) did not show changes when the athletes were supplemented with CrM plus HMB [36].

## 4. Discussion

The main purpose of this systematic review was to summarize all scientific evidence about the effect of CrM plus HMB supplementation on variables related to physical performance (strength performance [36,40], anaerobic performance [35,37,39], aerobic performance [39]), body composition (LBM, BM, FM, FFM) [36,37,40], markers of muscle damage (CK, LDH, LA) [35,36,37,39], and hormone status (testosterone and cortisol) [38], as measured in the six studies included in this systematic review [35,36,37,38,39,40]. The main results indicated that the combination of 3–10 g/day of CrM plus HMB 3 g/day, over 1–6 weeks, could produce more improvements than taking them in an isolated way in strength performance, anaerobic performance, and for 4 weeks in body composition (increasing FFM and decreasing FM). However, no significant results were found on aerobic performance, on markers of muscle damage, and on anabolic/catabolic hormone parameters when both supplements were combined. Due to the different measured outcomes in the studies, the following outcomes have been divided into different groups. The results could be influenced by type of sport, amount of each supplement, and duration of the intervention. Participant characteristics, such as age, gender, ethnicity, body composition, training level, differences in training, nutrition, and health status, may also influence the results.

### 4.1. Impact on Sport Performance

#### 4.1.1. Strength Performance

Strength is an essential attribute in sports performance, and describes the capacity to perform any action faster when the same mass is moved [45]. Strength tests require Cr/PCr as an energy substrate [46]. Therefore, strength could be enhanced with CrM ingestion, improving muscular performance by an increase of Cr/PCr [8]. In addition, HMB supplementation can also improve strength through an increase of the muscle cross-sectional area [29]. This effect of HMB could be due to an increase of muscle-protein synthesis caused by an up-regulation of the mTOR pathway [27], or by a marked change in oxidative metabolism [29]. Although in one of the two studies analyzed, the strength performance was improved [36]; in the other [40], no changes were found. The main reason for this could be the training status of the participants [47]. However, in the study by Jowko et al., 40 healthy males were measured by 1-RM of different exercises (bench press, behind the neck press, biceps curl, back squat, triceps extension, power clean, and sum of all tests), and it was shown that CrM plus HMB supplementation caused accumulative strength increases [36]. These observations are consistent with, but do not prove, the hypothesis that CrM plus HMB acts through distinct mechanisms, as already described.

#### 4.1.2. Anaerobic Performance

The prolonged periods of multiple sprints that occur in different sports (for example soccer, basketball, or rugby) drain muscle glycogen stores, leading to a decrease in energy production and a reduction in the overall work rate during training and/or competition [48] Short-term CrM supplementation has been shown to up-regulate the mRNA content of some genes and proteins involved in glycogen synthesis, producing a change in cellular osmolarity [15]. Moreover, an increase of muscle PCr by CrM supplementation is essential to activities dependent on the PCr energy system [13]. In addition, the Cr/PCr shuttle acts as a buffer that reduces LA in anaerobic glycolytic actions [16]. On the other hand, the negative muscle net balance that occurs after resistance exercise must be resolved quickly [49]. To achieve this, leucine must be ingested after exercise. Supplementation with HMB (leucine is a precursor of HMB) in the 30 minutes post the exercise period could be enough stimulus to produce protein synthesis, facilitating muscle recovery [29].

In relation to this topic, three studies were analyzed [35,37,39]. Two of them showed improvements in anaerobic performance [35,37]. Concretely, Faramarzi et al. [35] found a higher anaerobic performance (peak power) in soccer players during the RAST in CR/HMBG with respect to HMBG. This result showed an additive effect of combined CrM plus HMB supplementation [35]. Equally, Zajac et al. found, in elite basketballers, better statistical results in CRG and CR/HMBG compared to HMBG and CON in a triple Wingate test. However, although there were no statistical differences between CRG and CR/HMBG, CR/HMBG showed a better performance in a triple Wingate test [37]. On the other hand, O’Connor et al. did not find changes among supplemented groups after a 60 s maximal anaerobic capacity test [39]. These results could be related to the high-level anaerobic training status of elite rugby players, and the difficulty in improving results for such athletes [39]. Thus, the combination of CrM plus HMB could help in achieving better results in improving anaerobic performance than taking them individually, however more research is needed to affirm these findings.

#### 4.1.3. Aerobic Performance

Aerobic performance is a key factor in long endurance sports (for example long running, cycling, or rowing events) which require maintaining a specific intensity for as long as possible [50]. CrM can improve aerobic capacity by increasing the Cr/PCr shuttle that acts like a buffer, lowering the LA concentration at the same intensity [16]. In addition, CrM improves muscle glycogen synthesis [14,15]. HMB can also affect aerobic performance by enhancing aerobic capacity through mitochondrial biogenesis, by activation of PGC-1α [30]. In this regard, no significant differences were observed among supplemented groups in the only study where aerobic performance was measured by a multistage aerobic capacity test [39]. This result could be due to the fact that the training focused on resistance exercises. Furthermore, the participants were highly trained athletes, who would be closer to their maximum genetic potential compared to untrained subjects [47].

Therefore, these controversial results generate certain doubts about the supplementation of CrM plus HMB and the benefit to aerobic performance. However, future work should be oriented towards exploring their effects on other types of tests and among different groups of athletes.

### 4.2. Impact on Body Composition

To obtain maximum performance, athletes need to have an optimal body composition for the concrete sport practice, with a low-fat mass percentage and optimal skeletal muscle mass [51]. HMB could increase muscle mass [52] by augmenting protein synthesis after an intense training session [27], by activation of mTOR [53]. Equally, it can reduce protein breakdown through reducing the ubiquitin-proteasome system [54] and by increasing the GH-IGF-I axis [55]. In addition, HMB could augment fat oxidation, improving mitochondrial biogenesis by activation of PGC-1α, and thereby, lowering the fat mass percentage [30]. On the other hand, CrM supplementation has been proposed to increase muscle mass by increasing osmotic pressure in muscle, which increases the water content of the muscle [56,57,58], which in turn promotes glycogen synthesis [14,15].

The mixture of CrM plus HMB could augment FFM or LBM more than taking them in an isolated way, as the study by Zajac et al. [37] demonstrated. However, in the study by Jowko et al. [36], no changes were found. FM had also can be reduced [37] or unchanged [36]. In this sense, Zajac et al. [37] obtained an interesting result, given that mixed supplementation combined with three full-body resistance training sessions per week appeared to help to increase BM and decrease FM. HMBG decreased BF in comparison to CRG and CONG, and CRG increased BM (without increasing FM) in contrast to HMBG and CONG, showing an accumulative effect in terms of enhancing body composition when they were ingested together. Moreover, in the investigation by O’Connor et al. [40], body composition parameters measured (BM, sum of six skinfolds, arm girth (relaxed), arm girth (fixed), chest girth, waist girth, hip girth, and thigh girth) were not changed when combined with three full body resistance training sessions and one speed/power session per week. These results might be explained by the high training level of the participants. Therefore, the combination of CrM plus HMB could have an additive effect due to the performance of different energetic pathways.

### 4.3. Impact on markers of Muscle Damage and Hormone Status

#### 4.3.1. Markers of Exercise-Induced Muscle Damage

The markers of exercise-induced muscle damage markers that were measured, LDH, CK, and LA, could be predictors of training intensity. Therefore, the activity of these markers is potentially useful, not as a marker of impending overtraining, but as a means of identifying a state of recent muscle damage or temporary over-reaching [59]. When CrM and HMB are taken in an isolated manner CrM can reduce LA [60], LDH [12], and CK [12] levels, and HMB can also decrease LA [20], CK [61], and LDH [61] levels after training. On the other hand, when the supplements are taken together, they do not show positive effects on CK [35,36,38], LDH [35], and LA [37,39] levels. Concretely, in the study by Jowko et al. [36], CK levels remained elevated following three weeks of CrM/HMBG supplementation. This could explain a decrease of protein degradation via HMB [62], an increase of LBM and improvements in strength achieved by this supplementation. This study [36] did not show an additive effect of HMB and CrM; quite the opposite, with CrM impairing HMB’s results. In summary, the results of the studies showed no better effect from combining both supplements on markers of muscle damage markers compared to individual intake.

#### 4.3.2. Anabolic/Catabolic Hormones

Monitoring testosterone and cortisol could provide insight into an athlete’s recovery/readiness, and could be a tool to program daily volume/intensity of training [63]. While testosterone is an anabolic and anticatabolic hormone that indicates the degree of endogenous regeneration, cortisol indicates accumulated stress [63].

HMB ingestion can reduce blood cortisol levels [64] and CrM can increase testosterone levels [32,33] when they are taken individually. There was only one study that analyzed hormone status when the mixed supplements were ingested [38], and it showed no differences in cortisol or testosterone levels after six weeks of supplementation. It is difficult to obtain changes in this parameter in few time (6 weeks) [65], however, and more studies are needed to understand the effect of CrM and HMB in combination on anabolic/catabolic hormone responses.

### 4.4. Strengths, Limitations and Future Research

In this systematic review, we analyzed six studies [35,36,37,38,39,40] considering different sport modalities (rugby, basketball, and soccer), outcomes, and supplementation duration. Moreover, the tests used to measure strength, anaerobic performance, and aerobic performance were completely different. These differences resulted in difficulties in comparing the different outcomes of the studies. Furthermore, not all the studies analyzed the effect of the supplementation mixture on different groups. Therefore, the results of this systematic review should be treated with caution due to the small number of research works available to include that are relevant to this area. Accordingly, more studies with similar measurement methodologies are needed in order to determine the efficacy of mixing CrM plus HMB for improving sports performance, and in order to understand the potential additive effect of this combination.

## 5. Conclusions

In summary, the main results of this systematic review seem to indicate that the combination of 3–10 g/day of CrM plus 3 g/day HMB for 1–6 weeks could produce improvements in strength performance, anaerobic performance, and for 4 weeks in body composition (increasing FFM and decreasing FM), that may exceed the effects of taking them in an isolated way. However, no significant results relating to markers of exercise-induced muscle damage and anabolic/catabolic hormone status were found when both supplements were combined.

## Figures and Tables

**Figure 1 nutrients-11-02528-f001:**
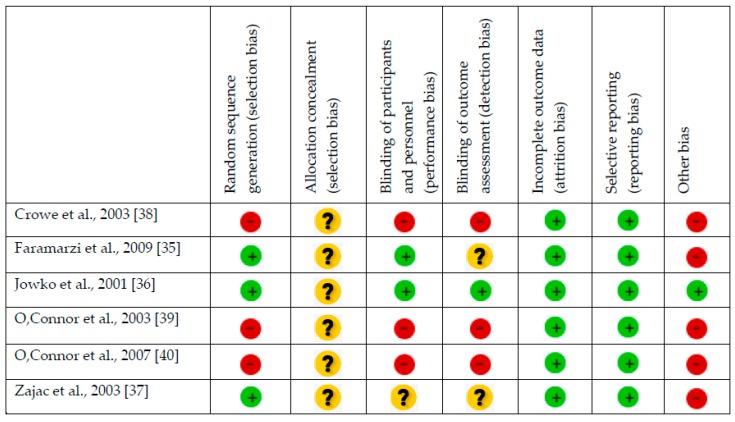
Risk of bias graph: review authors’ judgments about each risk of bias item presented as percentages across all studies. 
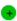
 Indicates low risk of bias; 
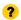
 indicates unknown risk of bias; 
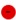
 indicates high risk of bias.

**Figure 2 nutrients-11-02528-f002:**
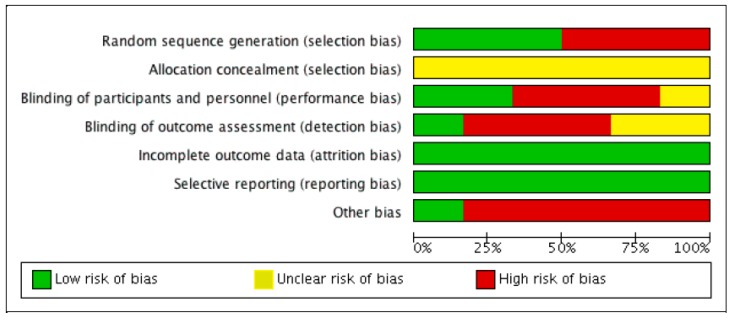
Risk of bias summary: review authors’ judgments about each risk of bias item for all studies.

**Figure 3 nutrients-11-02528-f003:**
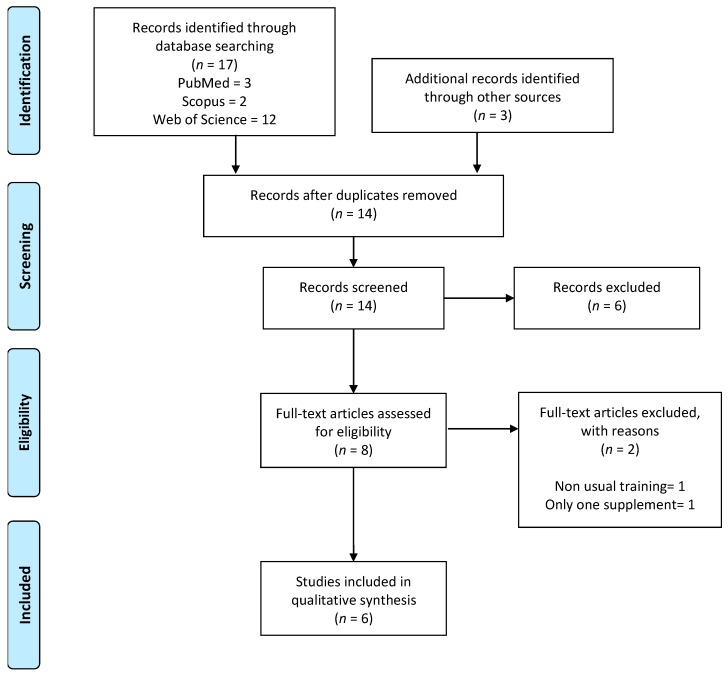
Preferred Reporting Items for Systematic Reviews and Meta-Analyses (PRISMA) flow diagram.

**Table 1 nutrients-11-02528-t001:** Summary of studies included in the systematic review that investigated the effect of CrM plus HMB on athletic performance abilities.

Author/s	Population	Intervention	Outcomes	Effects
CrM+HMB Vs CON/PLG	CrM+HMB Vs CrMG	CrM+HMB Vs HMBG
Faramarzi et al., (2009) [35]	24 soccer players(21.6 ± 0.1 years)	Randomized, placebo- controlledCrM: 3 g/dayHMB: 3 g/dayDuration: 6 days	Peak Power (RAST)Mean Power (RAST)Fatigue Index (RAST)	↑ Peak Power↑ Mean Power  Fatigue Index	No data shown	↑ Peak Power  Mean Power  Fatigue Index
Jówko et al., (2001) [36]	40 healthy males(21.0 ± 2.1 years)	Randomized, double-blind, placebo-controlledCrM: 20 g/day (first 1 week) + 10 g/day (2 weeks)HMB: 3 g/dayDuration: 3 weeks	Accumulative strength tests (1-RM)	↑ Accumulative strength tests (1-RM)	 Accumulative strength tests (1-RM)	 Accumulative strength tests (1-RM)
O’Connor & Crowe (2003) [39]	27 male elite rugby players(18–32 years)	ControlledCrM: 3 g/dayHMB: 3 g/dayDuration: 6 weeks	Aerobic performance (multistage aerobic capacity test)Anaerobic performance (60 second maximal anaerobic capacity test)	 Aerobic capacity  Anaerobic capacity	No data shown	 Aerobic capacity  Anaerobic capacity
O’Connor & Crowe (2007) [40]	30 male elite rugby players(24.9 ± 1.5 years)	ControlledCrM: 3 g/dayHMB: 3 g/dayDuration:6 weeks	Muscular strength (3RM test)Muscular endurance (maximum number of chin-ups to exhaustion)Peak power (Ten-second Leg Power Test)Total work (Ten-second Leg Power Test)	 Bench press  Deadlift  Prone row  Muscular endurance  Peak power  Total work	No data shown	 Bench press  Deadlift  Prone row  Muscular endurance  Peak power  Total work
Zajac et al., (2003) [37]	52 well trained basketball players(25.6 ± 5.6 years)	Randomized, placebo- controlledCrM: 15 g/day (first 5 days) + 5 g/day (rest of the days)HMB: 3 g/dayDuration: 30 days	Relative maximal anaerobic power (triple Wingate test)Relative total work (triple Wingate test)	↑ Relative maximal↑ Relative total work	 Relative maximal  Relative total work	↑ Relative maximal↑ Relative total work

CrM: Creatine monohydrate supplementation, HMB: HMB supplementation, CON/PLG: Placebo or control group, HMBG: HMB supplementation group, CrMG: Creatine monohydrate supplementation group, RAST: Running Anaerobic Speed Test; ↑: Increase, ↓: Decrease, 

: No effect.

**Table 2 nutrients-11-02528-t002:** Summary of studies included in the systematic review that investigated the effect of CrM plus HMB on body composition.

Author/s	Population	Intervention	Outcomes	Effects
CrM+HMB Vs CON/PLG	CrM+HMB Vs CrMG	CrM+HMB Vs HMBG
Jówko et al., (2001) [36]	40 healthy males(21.0 ± 2.1 years)	Randomized, double-blind, placebo-controlledCrM: 20 g/day (first 1 week) + 10 g/day (2 weeks)HMB: 3 g/dayDuration: 3 weeks	BMBFLBM	 BM  FM  LBM	 BM  FM  LBM	 BM  FM  LBM
O’Connor & Crowe (2007) [40]	30 male elite rugby players(24.9 ± 1.5 years)	ControlledCrM: 3 g/dayHMB: 3 g/dayDuration:6 weeks	BMSum of six skinfoldsArm girth (relaxed)Arm girth (fixed)Chest girthWaist girthHip girthThigh girthFemur diameterHumerus diameter	 BM  Sum of six skinfolds  Arm girth (relaxed)  Arm girth (fixed)  Chest girth  Waist girth  Hip girth  Thigh girth  Femur diameter  Humerus diameter	No data showed	 BM  Sum of six skinfolds  Arm girth (relaxed)  Arm girth (fixed)  Chest girth  Waist girth  Hip girth  Thigh girth  Femur diameter  Humerus diameter
Zajac et al., (2003) [37]	52 well trained basketball players(25.6 ± 5.6 years)	Randomized, placebo- controlledCrM: 15 g/day (first 5 days) + 5 g/day (rest of the days)HMB: 3 g/dayDuration: 30 days	BMFFMFM	↑ BM↑ FFM↓ FM	 BM  FFM↓ FM	↑ BM↑ FFM  FM

CrM: Creatine monohydrate supplementation, HMB: HMB supplementation, CON/PLG: Placebo or control group, HMBG: HMB supplementation group, CrMG: Creatine monohydrate supplementation group, LBM: Lean body mass, BM: Body mass, FM: Fat mass, FFM: Fat free mass; ↑: Increase, ↓: Decrease, 

: No effect.

**Table 3 nutrients-11-02528-t003:** Summary of studies included in the systematic review that investigated the effect of CrM plus HMB on markers of muscle damage and hormone status outcomes.

Author/s	Population	Intervention	Outcomes	Effects
CrM+HMB Vs CON/PLG	CrM+HMB Vs CrM	CrM+HMB Vs HMBG
Crowe et al., (2003) [38]	28 male elite rugby players(24.9 ± 0.7 years)	ControlledCrM: 3 g/dayHMB: 3 g/dayDuration: 6 weeks	TestosteroneCortisolCKUrea	 Testosterone  cortisol  CK  Urea	No data shown	 Testosterone  cortisol  CK  Urea
Faramarzi et al., (2009) [35]	24 soccer players(21.6 ± 0.1 years)	Randomized, placebo- controlledCrM: 3 g/dayHMB: 3 g/dayDuration: 6 days	CKLDH	 CK  LDH	No data shown	 CK  LDH
Jówko et al., (2001) [36]	40 healthy males(21.0 ± 2.1 years)	Randomized, double-blind, placebo-controlledCrM: 20 g/day (first 1 week) + 10 g/day (2 weeks)HMB: 3 g/dayDuration: 3 weeks	CKUrea nitrogen	 CK↓Urea nitrogen	↑CK  Urea nitrogen	↑CK  Urea nitrogen
O’Connor & Crowe (2003) [39]	27 male elite rugby players(18–32 years)	ControlledCrM: 3 g/dayHMB: 3 g/dayDuration: 6 weeks	LA (blood)	 LA	No data shown	 LA
Zajac et al., (2003) [37]	52 well trained basketball players(25.6 ± 5.6 years)	Randomized, placebo- controlledCrM: 15 g/day (first 5 days) + 5 g/day (rest of the days)HMB: 3 g/dayDuration: 30 days	LACKLDH	 LA  CK  LDH	 LA  CK  LDH	 LA↑ CK↑ LDH

CrM: Creatine monohydrate supplementation, HMB: HMB supplementation, CON/PLG: placebo or control group, HMBG: HMB supplementation group, CrMG: Creatine monohydrate supplementation group, CK: creatine kinase, LA: blood lactate, LDH: Lactate dehydrogenase; ↑: Increase, ↓: Decrease, 

: No effect.

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
