# Peer review of "Effect of the Combination of Creatine Monohydrate Plus HMB Supplementation on Sports Performance, Body Composition, Markers of Muscle Damage and Hormone Status: A Systematic Review"

_nutrients, 2019, doi:10.3390/nu11102528_

Round 1

Reviewer 1 Report

The authors have conducted a systematic review on the combined vs. isolated effects of creatine and HMB supplementation on sports performance, body composition and muscle recovery. The authors conclude that the combination of creatine and HMB over 1-6 weeks supplementation may enhance sport performance (strength and anaerobic performance), and body composition (fat free mass and fat mass). However, this combination did not show positive effects on markers of muscle recovery (anabolic-catabolic hormones and muscle damage). My comments appear below.

Major comments:

The quality of the written English requires substantial improvement. I suggest the authors have their manuscript edited by a professional science editor.

The systematic review is composed of only 6 research articles in total. Of that, only 5 for sports performance, 3 for body composition, and 5 for muscle recovery. In my opinion, there is not enough data to warrant a systematic review on these outcome measures.

2.4 and 3.3. These sections are redundant. The outcome measures (athletic performance, body composition, and muscle recovery) are not specifically stated in 2.4. Please state the specific outcomes examined.

Tables 1, 2, 3. These tables are not clear. Please revise.

Discussion: The Discussion does not really attempt to synthesize the findings of the different research studies. Instead, results from individual studies are often referred to. In my opinion, the authors need to do a better job on their Discussion in order to provide more insight to the reader. Do the authors recommend the use of both supplements? Recent evidence does not support the use of HMB to enhance lean body mass of strength with resistance exercise PMID:

30113522.

Minor comments:

Outcome measures: Please move the following to the Discussion section: “The results could be influenced by type of sport, amount of each supplementation and duration of the intervention. Participants characteristics, such as age, gender, ethnicity, body composition, training level, differences in training, nutrition and health status and ethnicity may also influence the results”.

How to the authors define exercise recovery? Are the hormones measured actually indicative of recovery? I do not think there is solid evidence to support this claim.

Author Response

The authors appreciate the time you devoted to reading our manuscript and helping us to craft an improved version of the investigation. We are pleased to clarify your concerns which we believe have improved the quality and applicability of your work. Please, find below our response to each of your observations. We have made a concerted attempt to systematically address the specific concerns raised for this revision and we have highlighted the alterations to this revision within the manuscript in red for your convenience.

Reviewer(s)' Comments to Author:

Major comments:

REVIEWER: The quality of the written English requires substantial improvement. I suggest the authors have their manuscript edited by a professional science editor.

AUTHORS: Thanks, so much for your help. The manuscript has undergone English language editing by MDPI. The text has been checked for correct use of grammar and common technical terms, and edited to a level suitable for reporting research in a scholarly journal. MDPI uses experienced, native English-speaking editors. Full details of the editing service can be found at https://www.mdpi.com/authors/english.

REVIEWER: The systematic review is composed of only 6 research articles in total. Of that, only 5 for sports performance, 3 for body composition, and 5 for muscle recovery. In my opinion, there is not enough data to warrant a systematic review on these outcome measures.

AUTHORS: Thank you very much for your comment. Indeed, in fact, there is a small number of papers in relation to this topic that could be a limitation of this systematic review and that the results obtained from it should be taken with caution. However, after reviewing some manuscripts that show the requirements for a high-quality systematic review, for the best of the authors´ knowledge, none of them refers to the minimum number of articles that should be included in it (1,2). However, these authors speak of strict quality criteria when preparing a systematic review (1,2). In this sense, this systematic review was carried out in accordance with PRISMA (Preferred Reporting Items for Systematic Reviews and Meta-analyses) statement guidelines and the PICOS model for the definition of the inclusion criteria. Moreover, methodological quality and risk of bias were assessed by two authors independently, and disagreements were resolved by third part evaluation, in accordance with the Cochrane Collaboration Guidelines samples.

However, assuming that possible limitation the authors have modified the conclusions, as well as adding a paragraph in this regard in the limitations, strengths and future lines of research section.

1- Charrois, T. L. (2015). Systematic reviews: what do you need to know to get started? The Canadian journal of hospital pharmacy, 68 (2), 144.

2- Garg, A. X., Hackam, D., & Tonelli, M. (2008). Systematic review and meta-analysis: when one study is just not enough. Clinical Journal of the American Society of Nephrology, 3 (1), 253-260

REVIEWER: 2.4 and 3.3. These sections are redundant. The outcome measures (athletic performance, body composition, and muscle recovery) are not specifically stated in 2.4. Please state the specific outcomes examined.

AUTHORS: Thank you very much for your recommendation. We agree. We have deleted the paragraph number 3.3.

REVIEWER: Tables 1, 2, 3. These tables are not clear. Please revise.

AUTHORS: Thanks, so much for this detail. We have changed the tables in order to be clearer.

REVIEWER: Discussion: The Discussion does not really attempt to synthesize the findings of the different research studies. Instead, results from individual studies are often referred to. In my opinion, the authors need to do a better job on their Discussion in order to provide more insight to the reader.

AUTHORS: Thank for your suggestion. The authors added at the of each discussion section. one sentence trying more insight to the reader.

REVIEWER: Do the authors recommend the use of both supplements?

AUTHORS: Thank you for your question. On the one hand, as already mentioned, the small number of articles reacted with this issue means that the results must be taken with caution. On the other hand, after complete this systematic review, the level of knowledge on this combination allows us open new investigation windows. The authors belong to a research group that works directly with athletes. A doctoral thesis is being carried out in relation to the effectiveness of supplementation with both supplements in aerobic power, as well as in muscle damage and anabolic and catabolic hormones. Preliminary results indicate that there is a synergistic effect of both supplements on aerobic power performance (the article is under review) and on the testosterone / cortisol index.

REVIEWER: Recent evidence does not support the use of HMB to enhance lean body mass of strength with resistance exercise PMID: 30113522.

AUTHORS: Thank you for your recommendation. The article referenced by the reviewer is of great interest to science as it demonstrates that HMB is no more effective in stimulating resistance training-induced hypertrophy and strength gains than leucine when is added to whey protein. In this sense, we must remember that HMB is a metabolite of leucine. However, we have understood the results presented do not compare with a placebo. This fact could generate caution when concluding that the HMB is not effective to achieve an increase in muscle mass and/or strength.

Minor comments:

REVIEWER: Outcome measures: Please move the following to the Discussion section: “The results could be influenced by type of sport, amount of each supplementation and duration of the intervention. Participants characteristics, such as age, gender, ethnicity, body composition, training level, differences in training, nutrition and health status and ethnicity may also influence the results”.

AUTHORS: Thank you for your recommendation. The authors have moved the paragraph to the first part of the discussion. And agree to this respect, given that is more appropriate after explain the main results inside the discussion part.

REVIEWER: How to the authors define exercise recovery? Are the hormones measured actually indicative of recovery? I do not think there is solid evidence to support this claim.

AUTHORS: Thank you for your comment. In order to avoid misunderstanding in this regard, the authors have modified the term muscle recovery due to muscle damage and hormone status. More specifically, muscle damage induced by exercise and state of anabolic catabolic hormones is indicated.

Reviewer 2 Report

Review of the article

Effect of combination of creatine monohydrate plus HMB supplementation on sports performance, body composition and muscle recovery: A systematic review(nutrients-605482)

The manuscript entitled “Effect of combination of creatine monohydrate plus HMB supplementation on sports performance, body composition and muscle recovery: A systematic review” (nutrients-605482) is well written, interestingly and presents valuable data.

This publication could have a valuable contributor to research on this type of topic. Simultaneously, it contains relatively few points which according to the reviewer opinion should be corrected before considering of its acceptance.

Individual specific comments are given below

Abstract and conclusions – authors wrote: “this combination did not show positive effects on muscle recovery (anabolic-catabolic hormones and muscle damage)” - I would suggest less resolute/strict conclusions - there are no enough of studies to confirm this sufficiently. It should be clearly stated that observations should be treated with caution due to the relatively small number of works in this area. Remove green underlines in keywords.

INTRODUCTION

Change “may enhance aerobic [3,4], anaerobic performance” to “may enhance aerobic [3,4] and anaerobic performance” Authors wrote: “HMB could enhance sports performance like aerobic [18–20], anaerobic” – but what aerobic, anaerobic? Capacity? Please clarify. Change “Mammalian Target of Rapamycin” to “Mammalian Target of Rapamycin kinase”. Authors wrote: “Moreover, this benefit in sports performance could be motivated by a augment lean body mass (LBM) [20,22,31]” - Make sure that Lean Body Mass or Fat Free Mass was analyzed in the cited studies and provide the correct indicator. If both should write both FFM and / or LBM. “In this sense, some studies showing possible improvements on performance [35–37], body composition[36,37]” – put the space “…composition [36,37]”.

3.3. Outcome measures

“at free mass (FFM) [36,37] and FM [36,37])and muscle recovery” – put the space “and FM [36,37]) and muscle recovery”.

RESULTS

TABLE 1 - The editorial issue of the tables should be thoroughly corrected - something has changed here. Please also move the name of this table - it is currently nested inside. TABLES - I suggest limiting the accuracy of data with one decimal place. Providing greater accuracy (e.g., ZajÄ…c et al) as to age or length can be misleading and inappropriate TABLES - Unify this abbreviation in the text and tables - You should be consistent for example: CrM or MCr or MCR. TABLES – Indicators “Effects Vs CON/PLG   Vs CRG           Vs HMBG” - this can be confusing – maybe ““Effects CrM+HMB vs CON/PLG    vs CRG            vs HMBG” TABLE 3 also have to be editorially improved for better readability. TABLE 3 correct the legend - "LA: lactic acid" to "LA: lactate’.

DISCUSSION

Remove colons e.g. “4.1.1. Strength performance:” or “1.2. Anaerobic performance:” The sentence: “In addition, the HMB also improve strength through an increase of muscle cross-sectional area [51] due to an increase of muscle-protein synthesis by an up-regulation of mTOR pathway [27], .” must be thoroughly reworded. There are here important In cited work [51] CrM not HMB was supplemented. This ref. 51 concerns work from Souza-Junior and collegues [2019] and therefore next part of this sentence (about HMB-induced muscle-protein synthesis [27] is inconsistent. Sentence: “an increase of muscle-protein synthesis by an up-regulation of mTOR pathway [27], .” -> the end of this sentence ",. " - should be improved editorially. Sentence: “and shown that CrMplus HMB supplementation caused accumulative strength increases [36].” -> put the space “CrM plus HMB” Authors wrote: “Moreover, an increase of muscle PCr by the CrM ingestion is essential to activities dependent of PCr [13]”. – this is unclear what PCr dependent? Something is missing here. Authors wrote: “In this sense, supplementation with HMB (Leucine is a precursor of HMB), in the 30 minutes post exercise period, could be enough stimulus to produce a protein synthesis that facilitates muscle recovery. [29].” - delete one dot “recovery [29].” The sentence “Concretely, Faramarzi et al. [34], presented in soccer players, presented higher anaerobic performance (peak power), during the running anaerobic speed test (RAST), in CR/HMBG respect HMBG.” must be thoroughly reworded. Firstly, the study of Faramarzi et al. is placed in the literature section not in position [34] but [35]. Secondly, this sentence is linguistically intricate ("presented in soccer players, presented higher anaerobic performance". The sentence: “Therefore, when these markers are high, the training time during the week must be reduced due the overtraining risk” is a bit too much simplification of the interpretation and practical application of these indicators. Please reword this statement. The sentence: “On the other hand, muscle damage markers: CK [35,36,38] and LDH [35] were significantly unchanged after mixed supplementation protocol like LA [37,39] levels” is unclear – please reword this. “HMBG decreased blood CK” – style/simplification. (PLACEBO/CONTROL) – maybe use abbreviation? “..with no significant statistical change)This affirmation could explain” – put the dot and space “change).This”.

Author Response

The authors appreciate the time you devoted to reading our manuscript and helping us to craft an improved version of the investigation. We are pleased to clarify your concerns which we believe have improved the quality and applicability of your work. Please, find below our response to each of your observations. We have made a concerted attempt to systematically address the specific concerns raised for this revision and we have highlighted the alterations to this revision within the manuscript in red for your convenience.

Reviewer(s)' Comments to Author:

The manuscript entitled “Effect of combination of creatine monohydrate plus HMB supplementation on sports performance, body composition and muscle recovery: A systematic review” (nutrients-605482) is well written, interestingly and presents valuable data.

This publication could have a valuable contributor to research on this type of topic. Simultaneously, it contains relatively few points which according to the reviewer opinion should be corrected before considering of its acceptance.

Individual specific comments are given below

REVIEWER: Abstract and conclusions – authors wrote: “this combination did not show positive effects on muscle recovery (anabolic-catabolic hormones and muscle damage)” - I would suggest less resolute/strict conclusions - there are no enough of studies to confirm this sufficiently. It should be clearly stated that observations should be treated with caution due to the relatively small number of works in this area. Remove green underlines in keywords.

AUTHORS: Thank you for your comment. Indeed, the few studies carried out in this area make it difficult to make a resounding conclusion. Therefore, in the conclusion of the abstract we have included this new sentence: “this combination seems not show positive effects on muscle recovery (anabolic-catabolic hormones and muscle damage).”

In the same way, a phrase has been included in the limitations, strengths and future research lines section indicating this limitation. The phase is as follows: “Therefore, the results of this systematic review should be treated with caution due to small number of Scientific articles in this topic.”

Finally, the authors have deleted the green color of the keywords section.

INTRODUCTION

REVIEWER: Change “may enhance aerobic [3,4], anaerobic performance” to “may enhance aerobic [3,4] and anaerobic performance”

AUTHORS: Thank you for your recommendation. The authors have done these changes.

REVIEWER: Authors wrote: “HMB could enhance sports performance like aerobic [18–20], anaerobic” – but what aerobic, anaerobic? Capacity? Please clarify.

AUTHORS: Thank you for your suggestion. The authors haver clarified that sentence: On the other hand, HMB could enhance sports performance like aerobic power and capacity [18–20], anaerobic capacity [18,20], strength [21–24], body composition [18,25], muscle damage [26] and hormone status [25].

REVIEWER: Change “Mammalian Target of Rapamycin” to “Mammalian Target of Rapamycin kinase”.

AUTHORS: Done.

REVIEWER: Authors wrote: “Moreover, this benefit in sports performance could be motivated by a augment lean body mass (LBM) [20,22,31]” - Make sure that Lean Body Mass or Fat Free Mass was analyzed in the cited studies and provide the correct indicator. If both should write both FFM and / or LBM.

AUTHORS: Thank you for your suggestion. Authors have included fat free mass in that sentence: “Moreover, this benefit on sports performance could be motivated by a augment lean body mass (LBM) and/or fat free mass (FFM)…“

REVIEWER: “In this sense, some studies showing possible improvements on performance [35–37], body composition[36,37]” – put the space “…composition [36,37]”.

AUTHORS: Thank you for your observation. The authors have added the space.

REVIEWER: 3.3. Outcome measures

REVIEWER: “at free mass (FFM) [36,37] and FM [36,37])and muscle recovery” – put the space “and FM [36,37]) and muscle recovery”.

AUTHORS: Thank you for your observation. The authors have added the space.

RESULTS

REVIEWER: TABLE 1 - The editorial issue of the tables should be thoroughly corrected - something has changed here. Please also move the name of this table - it is currently nested inside.

AUTHORS: Thank you for your observation. The authors have changed the table 1.

TABLES - I suggest limiting the accuracy of data with one decimal place. Providing greater accuracy (e.g., ZajÄ…c et al) as to age or length can be misleading and inappropriate

AUTHORS: Thank you for your observation. The authors have provided greater accuracy.

TABLES - Unify this abbreviation in the text and tables - You should be consistent for example: CrM or MCr or MCR.

AUTHORS: Thank you for your observation. The authors have unified this abbreviation.

REVIEWER: TABLES – Indicators “Effects Vs CON/PLG   Vs CRG           Vs HMBG” - this can be confusing – maybe ““Effects CrM+HMB vs CON/PLG    vs CRG            vs HMBG”

AUTHORS: Thank you for your recommendation. The authors have added these indicators.

TABLE 3 also has to be editorially improved for better readability.

AUTHORS: Thank you for your observation. The authors have modified the table 3.

TABLE 3 correct the legend - "LA: lactic acid" to "LA: lactate’.

AUTHORS: Thank you for your suggestion. The authors have changed LA: lactic acid" to "LA: lactate’.

DISCUSSION

REVIEWER: Remove colons e.g. “4.1.1. Strength performance:” or “1.2. Anaerobic performance:”

AUTHORS: Thank you for your suggestion. The authors have removed colons.

REVIEWER: The sentence: “In addition, the HMB also improve strength through an increase of muscle cross-sectional area [51] due to an increase of muscle-protein synthesis by an up-regulation of mTOR pathway [27], .” must be thoroughly reworded. There are here important In cited work [51] CrM not HMB was supplemented. This ref. 51 concerns work from Souza-Junior and collegues [2019] and therefore next part of this sentence (about HMB-induced muscle-protein synthesis [27] is inconsistent.

AUTHORS: Thank you for your observation. The reference 51 has been changed by reference 29 (Pinheiro, C. eta l., 2012) and the sentence has been thoroughly reworded: “In addition, the HMB supplementation also improve strength through an increase of muscle cross-sectional area [29]. This effect of HMB could be due to an increase of muscle-protein synthesis by an up-regulation of mTOR pathway [27] or by a marked change in oxidative metabolism [29].”

REVIEWER: Sentence: “an increase of muscle-protein synthesis by an up-regulation of mTOR pathway [27], .” -> the end of this sentence ",. " - should be improved editorially.

AUTHORS: Thank you for your recommendation. The authors have improved editorially the sentence: This effect of HMB could be due to an increase of muscle-protein synthesis by an up-regulation of mTOR pathway [27] or by a marked change in oxidative metabolism [29]”

REVIEWER: Sentence: “and shown that CrMplus HMB supplementation caused accumulative strength increases [36].” -> put the space “CrM plus HMB”

AUTHORS: Thank you for your observation. The authors have added space between CrM and plus.

REVIEWER: Authors wrote: “Moreover, an increase of muscle PCr by the CrM ingestion is essential to activities dependent of PCr [13]”. – this is unclear what PCr dependent? Something is missing here.

AUTHORS: Thank you for your observation. The authors have added energy system in this sentence: “Moreover, an increase of muscle PCr by the CrM supplementation is essential to activities dependent of PCr energy system [13].”

REVIEWER: Authors wrote: “In this sense, supplementation with HMB (Leucine is a precursor of HMB), in the 30 minutes post exercise period, could be enough stimulus to produce a protein synthesis that facilitates muscle recovery. [29].” - delete one dot “recovery [29].”

AUTHORS: Thank you for your observation. The authors have deleted one dot.

REVIEWER: The sentence “Concretely, Faramarzi et al. [34], presented in soccer players, presented higher anaerobic performance (peak power), during the running anaerobic speed test (RAST), in CR/HMBG respect HMBG.” must be thoroughly reworded. Firstly, the study of Faramarzi et al. is placed in the literature section not in position [34] but [35]. Secondly, this sentence is linguistically intricate ("presented in soccer players, presented higher anaerobic performance".

AUTHORS: Thank you for your observation: Authors have changed the Faramarzi et al. reference and have rewritten that sentence: “Concretely, Faramarzi et al. [35], presented higher anaerobic performance (peak power) in soccer players,….”

REVIEWER: The sentence: “Therefore, when these markers are high, the training time during the week must be reduced due the overtraining risk” is a bit too much simplification of the interpretation and practical application of these indicators. Please reword this statement.

AUTHORS: Thank you for your recommendation. The authors have changed this sentence: “Therefore, the activity of these markers is potentially useful, not as a marker of impending overtraining, but as a means of identifying a state of recent muscle damage or temporary over-reaching [63].”

REVIEWER: The sentence: “On the other hand, muscle damage markers: CK [35,36,38] and LDH [35] were significantly unchanged after mixed supplementation protocol like LA [37,39] levels” is unclear – please reword this.

AUTHORS: Thank you for your observation. The authors have changed this sentence by: “On the other hand, when the supplements were taken together, they did not show significant changes on CK [35,36,38], LDH [35], and LA [37,39] levels.”

REVIEWER: “HMBG decreased blood CK” – style/simplification. (PLACEBO/CONTROL) – maybe use abbreviation? “..with no significant statistical change)This affirmation could explain” – put the dot and space “change).This”.

AUTHORS: Thank you for your observations. The authors have corrected all of these issues.

Round 2

Reviewer 1 Report

Entire Manuscript

Change “exercise induced muscle damage” or “muscle damage” to “markers of exercise induced muscle damage” or “markers of muscle damage”. None of the cited studies directly measured muscle damage (i.e. ultrastructural damage via z-band streaming).

Abstract

“Third part” should be “third party”

When discussing body composition use “increases” or “decreases” not improvements since this is subjective.

How can fat mass or fat-free mass change within 1 week?

Author Response

We appreciate again the time you devoted to reading our manuscript and helping us to craft an improved version. We are pleased to clarify your concern which we believe will improve the impact and quality of your work. Please find below our response to your observation. We have made a concerted attempt to systematically address the specific concerns raised for this revision and we have highlighted the alterations to this revision within the manuscript in yellow for your convenience.

Reviewer(s)' Comments to Author:

Reviewer 1

REVIWER: Change “exercise induced muscle damage” or “muscle damage” to “markers of exercise induced muscle damage” or “markers of muscle damage”. None of the cited studies directly measured muscle damage (i.e. ultrastructural damage via z-band streaming).

AUTHORS: Thanks for your observation. Given that none of the cited studies directly measured muscle damage (i.e. ultrastructural damage via z-band streaming), the authors have changed “exercise induced muscle damage” or “muscle damage” to “markers of exercise induced muscle damage” or “markers of muscle damage”.

Abstract

REVIWER: “Third part” should be “third party”

AUTHORS: Thanks for your observation. The authors have changed “Third part” to “third party”.

REVIWER: When discussing body composition use “increases” or “decreases” not improvements since this is subjective.

AUTHORS: Thanks for your recommendation. When discussing body composition, the authors have changed “improvements” to use “increases” or “decreases”.

REVIWER: How can fat mass or fat-free mass change within 1 week?

AUTHORS: Thank you for your observation. The authors agree with the reviewer that is difficult change fat mass or fat-free mass within 1 week. In order to clarify this point, and based on the results of the articles reviewed, the authors have indicated that the combination of 3–10 g/day of CrM plus 3 g/day HMB for 1–6 weeks could produce improvements in strength performance, anaerobic performance, and for 4 weeks in body composition (increasing FFM and decreasing FM).
